



# High resolution aerosol data from the top 3.8 ka of the EGRIP ice core

Tobias Erhardt[1,2,*], Camilla M. Jensen[1,*], Florian Adolphi[1,2], Helle Astrid Kjær[3], Remi Dallmayr[2], Birthe Twarloh[2], Melanie Behrens[2], Motohiro Hirabayashi[4], Kaori Fukuda[4], Jun Ogata[4], François Burgay[5,6], Federico Scoto[5,7], Ilaria Crotti[5], Azzurra Spagnesi[5,8], Niccoló Maffezzoli[5,8], Delia Segato[5,8], Chiara Paleari[9], Florian Mekhaldi[9], Raimund Muscheler[9], Sophie Darfeuil[10], and Hubertus Fischer[1]

[1]Climate and Environmental Physics, Physics Institute, and Oeschger Center for Climate Change Research, University of Bern, Bern, Switzerland
[2]Alfred Wegener Institute, Helmholtz Center for Polar and Marine Science, Bremerhaven, Germany
[3]Physics of Ice, Climate and Earth, Niels Bohr Institute, University of Copenhagen, Copenhagen, Denmark
[4]National Institute of Polar Research, Research Organization of Information and Systems, Tokyo, Japan
[5]Ca' Foscari University of Venice, Department of Environmental Sciences, Informatics and Statistics, Venice, Italy
[6]Paul Scherrer Institute, Laboratory of Environmental Chemistry (LUC), Villigen PSI, Switzerland
[7]National Research Council of Italy, Institute of Atmospheric Sciences and Climate (ISAC-CNR), Lecce, Italy
[8]National Research Council of Italy, Institute of Polar Sciences, Venice, Italy
[9]Department of Geology, Lund University, Lund, Sweden
[10]University Grenoble Alpes, IRD, CNRS, INRAE, Grenoble INP, IGE, Grenoble, France
[*]These authors contributed equally to this work.

**Correspondence:** Tobias Erhardt (tobias.erhardt@awi.de)

**Abstract.** Here we present the high-resolution CFA data from the top 479 m of the East Greenland Ice coring Project (EGRIP) ice core covering the past 3.8 thousand years. The data consists of 1 mm-depth-resolution profiles of calcium, sodium, ammonium, nitrate and electrolytic conductivity as well as decadal averages of these profiles. Alongside the data we provide a description of the measurement setup, procedures, the relevant references for the specific methods as well as an assessment of

the precision of the measurements, the sample to depth assignment and the depth and temporal resolution of the data set. The nominally 1-mm data represents an oversampling of the record as the true resolution is limited by the analytical setup to approximately 1 cm. The error of absolute depth assignment of the data may be on the order of 1 cm, however relative depth offsets between the records of the individual species is only on the order of 1 mm. The presented data has sub-annual resolution over the entire depth range and has already formed part of the data for an annually layer-counted time scale for the EGRIP ice core used

to improve and revise the multi-core Greenland ice-core chronology (GICC05) to a new version,GICC21 (Sinnl et al., 2021). The data is available in full 1-mm resolution and decadal averages on PANGAEA (https://doi.org/10.1594/PANGAEA.945293 (Erhardt et al., 2022b))



## 1 Introduction

Ice cores from polar regions and their proxy records have allowed us to study the past climate and its variability in great detail.

Concentrations of aerosol constituents in the ice are among the widely used ice-core proxy records. Deposited onto the ice sheet they are well preserved in the ice matrix as either soluble or insoluble impurities accessible through various analysis techniques. They reflect the past aerosol deposition to the ice sheets at high temporal resolution, often allowing us to resolve the seasonal variability of the aerosols species if measured at high enough depth resolution. The high temporal resolution in turn can enable the generation of annual-layer counted age models from the aerosol records (e.g. Sinnl et al., 2021). Aerosol records of both

shallow and deep ice cores are routinely measured at high depth resolution using continuous melting and analysis setups, i.e., using so-called continuous flow analysis (CFA) (e.g. Sigg et al., 1994; Kaufmann et al., 2008). These setups have overcome some of the analytical challenges/limitations that the very low impurity concentrations and the desired high depth resolution pose and have become the gold standard for long, high-resolution aerosol records.

Typically, ice cores that aim to provide climate reconstructions far back in time are drilled away from fast-flowing ice, ideally

on the domes and ice-divides of an ice sheet, where effects of glacial flow on the records are minimal. The East Greenland ice coring project ice core (EGRIP) with its location inside of the North East Greenland ice stream (NEGIS) (Vallelonga et al., 2014; Hvidberg et al., 2020) is far from this ideal. However, the goal of this project was not only a long climate record, covering the Holocene and the Glacial Termination, but rather to the study of the NEGIS itself, including its causes and temporal changes. The high-resolution chemical impurity records from the EGRIP ice core provide not only a climate record,

but allow the detailed study of the interactions between impurities, micro-structure and ice rheology in this unique glaciological setting (Stoll et al., 2021, 2022).

In this paper we present data from the Bern wet-chemistry CFA, covering $Ca^{2+}$ (calcium), $NH_4^+$ (ammonium) and $NO_3^-$ (nitrate) concentrations, as well as electrolytic meltwater conductivity. Furthermore we provide data from the online ICP-TOFMS measurements of $Ca$ and $Na$ (sodium) concentrations. Along with the data sets, we describe the changes made to the

Bern CFA system for the EGRIP CFA campaigns since the last major measurement campaigns (Kaufmann et al., 2008; Erhardt et al., 2022a). These consist of the addition of an inductively coupled plasma time-of-flight mass spectrometer (ICP-TOFMS) to the system (Erhardt et al., 2019) as well as updated melting and calibration procedures for the wet-chemistry CFA system. We also provide a discussion of the uncertainties related to these changes covering analytical precision, depth-assignment and resolution. This paper accompanies the publication of the revision of the Greenland ice core chronology (GICC21) presented

in Sinnl et al. (2021), that makes extensive use of the data sets shown and discussed here.

## 2 Coring location

The EGRIP drill camp is located 359 km NNE of the Greenland Summit at 75.63° N, 36.00° W and an altitude of 2708 m a.s.l. as shown in Figure 1. Current accumulation rates at EGRIP are around 11 cm a$^{-1}$ (Vallelonga et al., 2014). Drilling at the site commenced in 2016 and has reached a depth of 2121 m in 2019 when the drilling was interrupted due to technical difficulties

and had to be postponed in the following seasons due to the COVID-19 pandemic. The site is located within NEGIS, 150 km



down-stream of the ice divide. Horizontal surface velocity at the coring location is around $55\,\mathrm{m\,a^{-1}}$ as determined by GPS measurements between 2015–2019 (Hvidberg et al., 2020). For all the data from the EGRIP ice core, including the data presented here, the coring location in the ice stream means, that going back further in time, the snow that formed the archive was deposited further upstream, at higher elevation and closer to the ice divide. At 3.8 ka, the ice likely originated approximately

111 km further upstream at an elevation about 160 m higher than the coring location and closer to the ice divide (Gerber et al., 2021). This area is characterized by a slightly higher accumulation rate of around $15\,\mathrm{cm\,a^{-1}}$. This upstream accumulation rate increase balances out the down-core thinning of the annual layers, leading to almost constant annual layer thickness over large parts of the Holocene (Gerber et al., 2021; Mojtabavi et al., 2020).

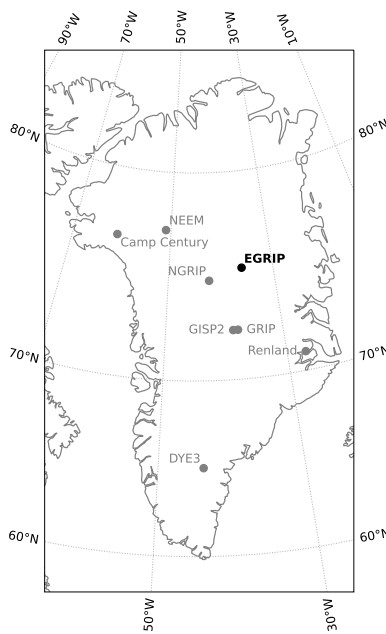

**Figure 1.** Coring location of the EGRIP ice core in bold with other major ice core drilling sites in Greenland

## 3  Sample description

For the continuous flow analysis measurements, samples with a cross section of 36 mm by 36 mm and a length of 0.55m were melted on a gold-plated melthead and only the innermost 26 mm by 26 mm area was used to supply the clean meltwater stream (Bigler et al., 2011). The measurements of the data down to 479 m depth presented here were performed in two melting campaigns in 2018 and 2019 covering the depth range of 13.75–350.35 m and 350.35–904.75 m respectively. The samples for the measurements down to 350.35 m were entirely cut in the field and 0.55 m pieces were than shipped via Copenhagen to

Bern for analysis. Below that, the chemistry CFA sticks and the neighboring gas CFA sticks were not split in the field but were shipped as 36 mm by 72 mm by 0.55 m slabs to Copenhagen, where they were split prior to the shipment to Bern. This was



done to guarantee higher sample quality for the CFA measurements in the brittle-ice section of the core, starting below the depth section presented here.

## 4    EGRIP continuous flow analysis

All measurements were performed at the continuous flow analysis lab at the University of Bern, instead of in the field as had been the case for the NGRIP and NEEM deep-drilling campaigns in Greenland. This change was necessary due to the addition of an ICP-TOFMS to the setup that is not easily field-deployable. In the CFA lab, the melthead is located in a -20 °C cold cell that is also used to prepare the samples for the measurements. The meltwater is pumped to the warm part of the lab for subsequent analysis.

For the EGRIP measurement campaigns, the analytical methods of the Bern CFA system were used basically unchanged from the NEEM (North Greenland Eemian Ice drilling, Dahl-Jensen et al. (2002)) system (Kaufmann et al., 2008; Erhardt et al., 2022a) with the exception of some changes to the melting procedures and the addition of a directly coupled ICP-TOFMS (icpTOF R, TOFWERK, Thun, Switzerland) (Erhardt et al., 2019). The addition of an ICP-TOFMS constitutes the largest change to the previously described system and will be outlined separately in Section 4.2.

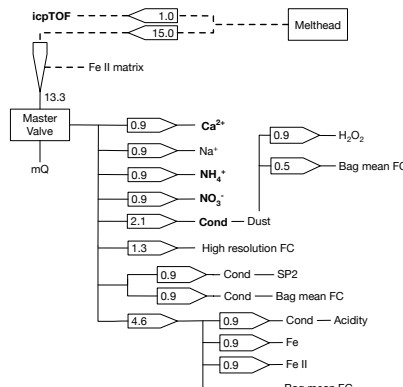

**Figure 2.** Water flow distribution from the central part of the melthead between all channels in the EGRIP CFA setup used for the top 350 m of the core in the 2018 melt campaign. Not shown is the meltwater stream from the outside of the core sample that was collected for cosmogenic radionuclide measurements. All arrows denote peristaltic pump channels with pump rates given in $\mathrm{ml\,min^{-1}}$. Dashed connections denote mixed air/water streams, solid lines pure water streams. Bold labels indicate channels used for the data presented here. Bag mean FC denotes fraction collection (FC) at 0.55 m resolution.

In addition to the measurements described here, the CFA system supplied clean meltwater streams to additional analysis channels and sampling efforts. The complete distribution of the meltwater streams between the different analysis channels of the Bern system and the measurements of sampling efforts of the partner institutes for the 2018 melting campaign are shown in Figure 2. These were setup and run in Bern by the international partners in the EGRIP project and include the continuous analysis of dissolved iron (Fe) and Fe II (Burgay et al. (2018), run by the University of Venice, Italy), Black Carbon using an



SP2 instrument (Mori et al. (2016), run by the National Institute for Polar Research, Japan) and an absorption spectroscopic measurement of the meltwater acidity (Kjær et al. (2016), run by the Center for Ice and Climate, Copenhagen, Denmark). The acidity measurements were integrated into the Bern CFA system in 2019 (for data starting from $350.35\,\mathrm{m}$) using a purpose-built spectrometer similar to the once used for all other channels in the Bern system. Furthermore, samples at $5--2.5\,\mathrm{cm}$ depth resolution for ion chromatography (run by the Alfred Wegener Institute, Bremerhaven, Germany) as well as multiple aliquotes

at 55-cm resolution were collected for various offline measurements run by the international partners. Finally, excess meltwater from the outer part of the ice stick was collected in $50\,\mathrm{ml}$ aliquotes for high-resolution measurements of cosmogenic $^{10}\mathrm{Be}$ and $^{36}\mathrm{Cl}$ (Paleari et al., 2022a, b).

### 4.1 Wet chemistry measurements

In the following section we provide the main references that describe the system and refer the reader to Erhardt et al. (2022a)

for a more detailed discussion of the system and its development over the years. The wet-chemistry (or main Bern CFA) system uses a range of analyte specific absorption and fluorescence spectroscopy methods to measure the low concentration of dissolved $\mathrm{Ca}^{2+}$ (Tsien et al., 1982), $\mathrm{Na}^+$ (Quiles et al., 1993), $\mathrm{NH}_4^+$ (Genfa and Dasgupta, 1989), $\mathrm{NO}_3^-$ (McCormack et al., 1994) and $\mathrm{H}_2\mathrm{O}_2$(Dasgupta and Hwang, 1985) (hydrogen peroxide) in the ice. All of these methods were specifically adapted for the CFA ice-core measurements (Sigg et al., 1994; Röthlisberger et al., 2000; Kaufmann et al., 2008) and employ custom-

built spectrometers as described in detail in (Kaufmann et al., 2008). Furthermore, the system employs a conductivity cell (Amber Science) to determine the electrolytic meltwater conductivity (Cond.) and a laser attenuation particle counter and sizer (Abalus, Klotz GmbH) for insoluble particle measurements. In 2019 (starting from $350.35\,\mathrm{m}$) the absorption spectroscopic $\mathrm{Na}^+$ measurement was dropped in favor of the more reliable and higher resolution measurement by the ICP-TOFMS (Erhardt et al., 2019). Furthermore, $\mathrm{H}_2\mathrm{O}_2$ measurements were halted after the 2018 season. Here we only provide data of $\mathrm{Ca}^{2+}$, $\mathrm{NH}_4^+$

and $\mathrm{NO}_3^-$ concentrations, electrolytic meltwater conductivity and ICP-TOFMS measurements of $\mathrm{Ca}$ and $\mathrm{Na}$ concentrations.

### 4.2 Continuous ICP-TOFMS

The addition of an inductively coupled plasma time-of-flight mass spectrometry (ICP-TOFMS) is the biggest analytical change of the Bern CFA system in comparison to previous melting campaigns. Coupling and integration of the ICP-TOFMS to the Bern CFA setup and the instrument parameters have been described in detail in Erhardt et al. (2019) and will only briefly be

outlined in the following. The main feature of the icpTOF is the fact that the time-of-flight mass spectrometer can measure the complete mass range of m/z 23 to 275 at a sampling rate of $33\,\mathrm{kHz}$ allowing the temporal resolution of short transient signals and coverage of almost the entire table of elements. For the CFA analysis, the instrument is run in two different data acquisition modes: either at an integration time of $250\,\mathrm{ms}$ for normal data acquisition of total elemental concentrations and at $2\,\mathrm{ms}$ integration time to also resolve the ionization events of individual particles entering the plasma in the single-particle

mode.

The sample stream for the ICP-TOFMS is diverted close to the melthead from the main meltwater stream. The air-water mixture coming from the melthead is used to efficiently transport particulates to the instrument and to limit signal dispersion.



Note, that the air is released from the ice during melting and thus does not pose a contamination risk. The air is only removed very close to the instrument using a custom build Teflon micro-volume debubbler with a volume of approximately $100\,\mu$l.
The bubble-free sample stream is then slightly acidified to $1\,\%$ using high-purity nitric acid (Optima Grade, Fisher Scientific) close to the sample introduction of the instrument. In this way washout times from the nebulizer and spray chamber can be minimized without risking the dissolution of particles in the ice which are the focus of the single-particle measurements. During the acidification, Rh (rhodium) is added to the sample as internal standard to monitor system stability and correct for sensitivity drifts. The sample is introduced into the plasma via a quartz cyclonic spray chamber at a flow rate of $400\,\mu$l min$^{-1}$
using a glass concentric nebulizer (Micro Mist, Glass Expansion) for the 2018 melting campaign (to $350\,$m depth) and a Teflon nebulizer (MicroFlow PFA-ST, Elemental Scientfic) thereafter. Detailed information on plasma conditions used for both cases can be found in (Erhardt et al., 2019).

Before and after each measurement run the instrument is calibrated with a range of multi-element standards. All standards are prepared by gravimetrically mixing multi-element standards (Sigma Aldrich, Inorganic Ventures) for a stock solution which
is then further diluted into working standards that span the expected range of concentrations in the ice. Due to the wide mass range that is covered by the TOFMS and the fact that now mass-scanning is necessary to acquire different masses, the range of analytes that can be targeted is basically only limited by their calibration and their possible interference. Even though the data presented here only contains Na and Ca concentrations, the instrument is routinely calibrated for a wide range of analytes including halogens, rare earth elements as well as heavy metals and platinum group elements. These data are still
under evaluation and are not provided here. For the calibration measurements a SC-$\mu$DX auto sampler with FAST injection valve (Elemental Scientific) was used to allow for repeatable and time efficient calibrations. Measurements were performed by injection of a $400\,\mu$l loop into a dilute nitric acid carrier stream which was run continuously between measurement runs.

For the data presented in this study, both normal ($4\,$Hz) and single-particle ($500\,$Hz) acquisition data were averaged to the nominal 1-mm resolution used for all of the data series from the main CFA system. The final data set is then smoothed with
a $10\,$s cutoff frequency Gaussian filter to remove high-frequency measurement noise from the data set. The depth assignment of the ICP-TOFMS data was achieved by aligning the Ca data to the raw fluorimetric Ca$^{2+}$ measurements of the main CFA system and thus to its depth-scale.

Figure 3 shows an overview of all data sets provided here at a $10\,$yr resolution on the GICC21 (Sinnl et al., 2021) age scale, both from the wet-chemistry CFA as well as from the ICP-TOFMS. Note, that even though the final data set of Ca$^{2+}$ exhibits
large gaps, the depth assignment of the ICP-TOFMS Ca data set is done using the raw fluorescence intensity data, as elaborated above. Thus the depth assignment of the Ca data is well constrained even when calibration issues affects the Ca$^{2+}$ data.

## 4.3 Updated melting and measurement procedure

Previously, the Bern CFA system was run with $1.10\,$m to $1.65\,$m of ice per run at most (Erhardt et al., 2022a), i.e. 2–3 times $0.55\,$m per continuous measurement run. This maximum length is dictated by the length of the frames used to guide the ice
during melting and by the vertical space available for the melting unit. Between each of the measurement runs, all connected instruments are run on an ultra pure water stream. Depending on the method it can take a significant time to reach a stable



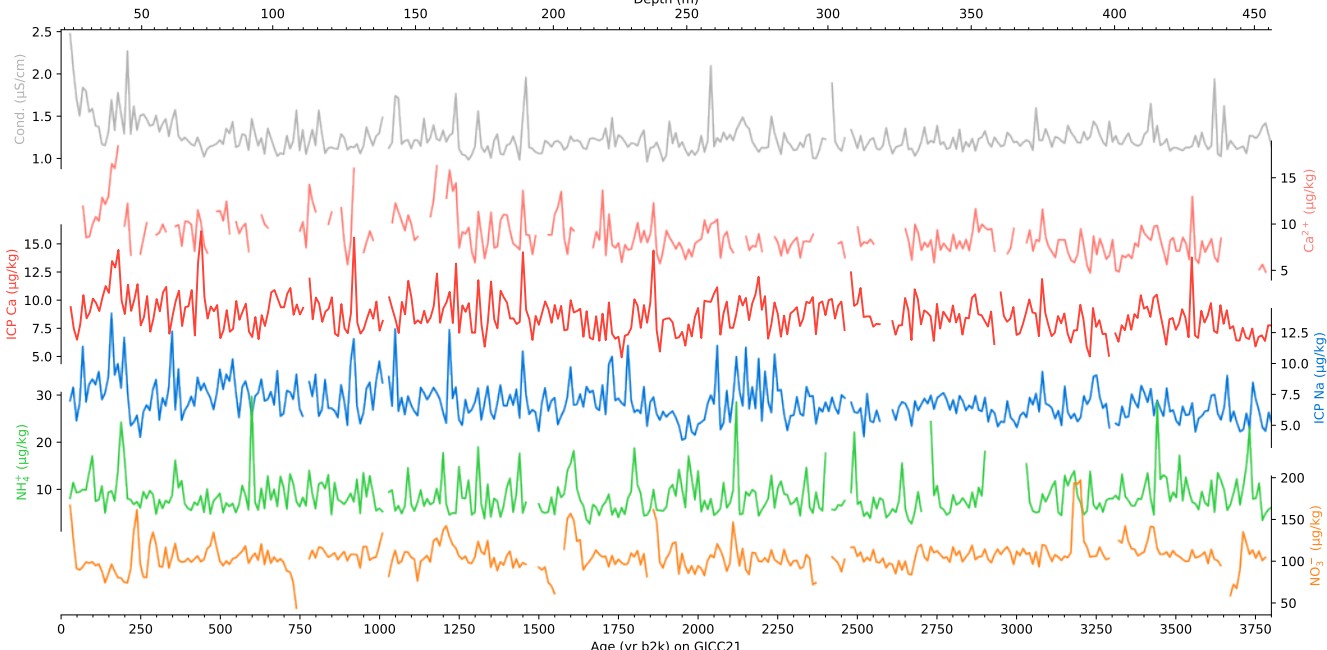

**Figure 3.** Overview of the complete data set at 10 yr resolution on the GICC21 (Sinnl et al., 2021) agescale. From top to bottom the data series are electrolytic meltwater conductivity (Cond.), $Ca^{2+}$ measured by fluorescence spectrometry, Ca and Na measured by ICP-TOFMS, $NH_4^+$ measured by fluorescence spectrometry and $NO_3^-$, measured by absorption spectrometry. Note, that the $Ca^{2+}$ record partly suffers from analytical issues leading to a large number of gaps in the final data set. However, the Ca data from the ICP-TOFMS covers mot of these gaps and provides an almost complete data set.

baseline signal between the runs, which is needed to catch and correct for drift in the analysis units. In a typical campaign, 2 to 3 runs were performed before a calibration run approximately every two to three hours which means that the waiting time between each run adds up to a large fraction of each measurement day.

To capitalize on the stability of the well matured Bern CFA system and the environmentally more stable lab setting as compared to the field, the melting procedure for the EGRIP ice core was changed to allow the uninterrupted measurement of multiple frames worth of ice-core samples by exchanging the frames mid-run. This is achieved by adding an additional fixed sample guide shown in Figure 4 to the Teflon melthead cover similar to what is also done in other CFA labs. During a measurement run, the frames can then be exchanged as soon as there is only so much ice left on the melthead ($\sim 10\,\mathrm{cm}$) that

it reaches the top of the new sample guide. To do so, the weight connected to the depth encoder is lifted from the remaining ice, the empty frame is removed from the holder and a new one is put in its place. The new piece of ice is then lowered carefully down to the remainder on the melthead using a Teflon wedge. Finally the encoder weight is then placed on top of the new sample. After some training, the whole procedure can be done within a few seconds during which the melting of the small remainder of ice continues. In this time a little less meltwater is produced, however the amount is still enough to avoid





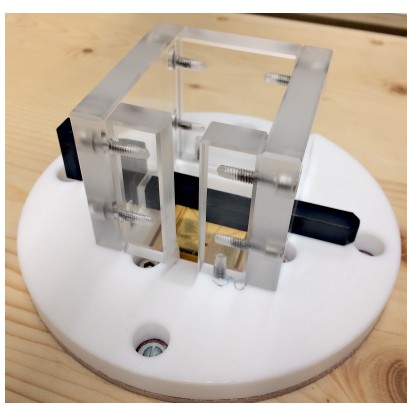

**Figure 4.** Picture of the melthead and the additional transparent sample guide used for the stacking of core segments during melting.

contamination by inflow from meltwater from the outside of the sample. During the procedure no depth recording is possible, however due to the short amount of time needed to re-position the encoder on the new piece of ice only very little ice melts during this time, as discussed below. The total amount of ice that can be measured in each measurement run is limited by the system stability and the frequency of calibration runs between extended ice measurements.

For the data presented here, usually a total of $3.30\,\mathrm{m}$ of ice were melted within each run, amounting to approximately two hours of uninterrupted measurement time per run at a meltspeed of $2.7\,\mathrm{cm\,min^{-1}}$. During each measurement day, a total of four runs with bracketing calibrations were performed. Depending on sample quality, this amounts to a total daily production of approximately $12$–$14\,\mathrm{m}$ per $\sim 16$ hour workday including the daily startup and shutdown of all instruments.

## 5 Data quality

The discussion of the uncertainties is split into three major parts: The analytical precision of the measurements, the depth assignment and the resulting depth and temporal resolution. To complete the discussion we added a few notes on the effect of the meltspeed stability on the data resolution as well as a brief comparison of the to calcium data sets.

### 5.1 Analytical precision

Because the spectroscopic methods used here were essentially not modified since the last measurement campaign (i.e. NEEM), the detection limits and uncertainties are assumed to have remained largely unchanged. The relevant numbers for each of the detection channels are listed in Table 1. As discussed in Gfeller et al. (2014), the uncertainty in the wet-chemistry is dominated by the uncertainty of the standard preparation by use of the water dispenser and microliter-pipette. For the concentration ranges discussed here, these uncertainties typically amount to less then $10\,\%$. For the wet-chemistry CFA data, the limit of detection (LOD) was determined as three times the standard deviation of the baseline/blank signal from ultra pure water (mQ, Merck/Millipore). In the case of the $\mathrm{Ca^{2+}}$ data, an issue with the fluorescence detector caused a low sensitivity in the top



322.3 m of the data set leading to a much higher limit of detection in this section of the core as indicated by the two different values in Table 1. The lower sensitivity and the resulting higher limit of detection also affects the $Ca^{2+}$ concentration range for the upper part of the core as values below the LOD were removed from the data.

**Table 1.** Analytical figures of merit for the wet-chemistry data. The sample concentration ranges are given by the 5 th and 95 th percentile and the median value in parentheses. The limit of detection is calculated as three times the standard deviation of the blank signal. The temporal resolution of the detection channels is determined as the long-term average 10–90 % rise time which is converted to an equivalent depth resolution assuming a melt speed of 2.7 cm min$^{-1}$. Note that this resolution does not account for signal dispersion during melting and in the debubbler.

| | Range (ppb) | LOD (ppb) | $t_{10-90}$ (s) | Res. (cm) |
|---|---|---|---|---|
| $Ca^{2+}$ (<322.3 m) | 2.3–23.3 (6.8) | 2.2 | 22.9 | 1.06 |
| $Ca^{2+}$ (>322.3 m) | 1.5–20.1 (5.4) | 0.2 | 22.9 | 1.06 |
| $NH_4^+$ | 0.8–28.0 (3.7) | 0.3 | 22.1 | 1.00 |
| $NO_3^-$ | 61.8–162.4 (99.1) | 1.2 | 26.9 | 1.21 |
| Cond. | 0.88–1.82 (1.16) | — | 20.6 | 0.93 |

For the ICP-TOFMS measurements of Ca and Na, the relevant performance numbers are given in Table 2. Here the values include the relative uncertainties from the standard errors of the calibrations. In the case of the ICP-TOFMS data, LODs were

determined using three times the standard error of the regression for the calibrations. These regressions include the blank measurements and allow for a non-zero intercept. This approach was chosen as it yielded more reliable results and allows for the determination of the LODs even in the occasional case if no reliable blank measurement could be performed.

**Table 2.** Analytical figures of merit for the ICP-TOFMS data. The sample concentration ranges are given by the 5 th and 95 th percentile and the median value in parentheses. The relative uncertainty (Rel. Uncert.) values are given for the respective concentrations in the range column. The limit of detection is determined from three times the standard error of the regression line at its intercept. The temporal resolution of the detection channels is determined as the long-term average 10–90 % rise time which is converted to an equivalent depth resolution assuming a melt speed of 2.7 cm min$^{-1}$. Note that this resolution does not account for signal dispersion during melting and in the debubbler.

| | Range (ppb) | Rel. Uncert. (%) | LOD (ppb) | $t_{10-90}$ (s) | Res. (cm) |
|---|---|---|---|---|---|
| Ca | 1.8–23.9 (5.8) | 6.6–3.6 (3.8) | 0.28 | 20.5 | 0.92 |
| Na | 0.4–22.3 (3.9) | 14.1–1.3 (1.9) | 0.17 | 17.9 | 0.81 |

## 5.2   Depth assignment

Previous investigations (Röthlisberger et al., 2000; Hiscock et al., 2013), have reported the uncertainty of the depth assignment

of the 110–165 cm measurements runs as ±1 cm. The total precision of the depth assignment is likely limited by the precision



of the length measurements during sample preparation due to possible influence of parallax effects and the limited precision of rulers (Erhardt et al., 2022a). However, due to the aforementioned change in the melt protocol in comparison to previous studies the quality of the depth assignment merits some additional discussion below. Overall, the uncertainty of the absolute depth assignment of all records to the ice-core depth is about $1\,\mathrm{cm}$, while the offsets between the individual CFA records is

only on the order of $1\,\mathrm{mm}$.

As mentioned above, during the stacking, the weight that is connected to the depth encoder is lifted from the ice. During that time, the remaining $\sim 10\,\mathrm{cm}$ of ice slowly continues to melt, albeit slower than usual, due to the missing pressure from the encoder weight. Due to the encoder being removed during that time, usually for around 5–15 s, the progress of the melting is unrecorded. The resulting gaps in the depth assignment are filled during the data processing using linear interpolation.

Additionally, because the amount of ice is greater per run, the total amount of ice contains more breaks, each of which carries the risk of adding to the depth-assignment uncertainty due to fact that each break surface will need to be cleaned and squared, leading to possible miss-measurement of the total length of ice.

To check whether the changed procedure introduces an additional depth-assignment error, we look at the total discrepancy between the length of ice as measured by the encoder to the length of ice measured using sample preparation. The relative

deviation of these values provides a measure of the accumulated uncertainty in the depth assignment at the end of the run. This relative deviation is shown in Figure 5G alongside other variables that describe the melting process. Note, however, that these deviations include the uncertainty of the length measurements during sample preparation and only show their discrepancy to the length measured by the encoder. Thus, they only provide a measure of the combined uncertainty.

Comparing the deviations to the length-statistics shown in Panels A-C of Figure 5, most notable, the relative deviation

between melted and measured sample lengths for the runs does not show any correlation with the total amount of ice melted or the number of pieces that make up the total length of the ice. Overall, we judge the relative uncertainty of the depth assignment not to be affected by the core-stacking process beyond the approximately $1\,\%$ previously reported for solid ice (Erhardt et al., 2022a). However, in the shallow part of the ice core, above the firn/ice transition where the melt speed is significantly higher than $2.7\,\mathrm{cm\,min^{-1}}$, the melted length of ice is systematically lower than the amount measured before the melting. We suggest,

this can be attributed to a slight compaction of the fragile firn during sample handling and the upright melting due to the overburden of the sample and the encoder weight. The change in melting procedure has not increased the uncertainty in the depth assignment beyond the approximately $1\,\mathrm{cm}$ in solid ice that were previously reported. As discussed in Erhardt et al. (2022a) this uncertainty mainly arises from the sample preparation, and only to a small part from the melting itself.

It is worth noting however, that these sources of uncertainty for the absolute depth assignment do not contribute to the phasing

error between the different analytes. The relative alignment between the parameters of the main CFA system is performed as described previously using short injections of a multi-component standard to determine the delays to each of the wet-chemistry channels. As discussed in (Erhardt et al., 2022a), the uncertainty of this relative phasing is on the order of 1–2 s, translating to less than a millimeter at the melt speeds employed here. The relative alignment of the ICP-TOFMS data is performed by alignment of the $Ca$ to the $Ca^{2+}$, transferring the relative phasing uncertainty of the wet-chemistry CFA to the ICP-TOFMS





measurements (Erhardt et al., 2019). In summary, while the uncertainty in absolute depth of all records may be about $1\,cm$ the relative depth offsets between individual CFA records is only on the order of $1\,mm$.

## 5.3   Depth and temporal resolution

The resolution for both the wet-chemistry CFA data sets as well as the ICP-TOFMS was determined by the 10–90 % rise time of standard measurements, treated in the same way as the sample measurements with respect to temporal resolution and
smoothing of the raw signals. Overall, the resolution of the data is on the order of approximately $1\,cm$ as listed in Tables 1 and 2. The $NO_3^-$ data, due to the long tubing and the reaction column involved, exhibits a slightly lower resolution at $1.2\,cm$. Both the electrolytic meltwater conductivity and the ICP-TOFMS measurements have slightly better resolution, below $1\,cm$. Note, that these values do not include the additional dispersion that happens during melting and in the debubbling volumes. However, as argued before (Erhardt et al., 2022a; Sigg et al., 1994), the response time of the detection usually dominates the overall
signal dispersion and the numbers reported can serve as a good guidance for the overall resolution of the data. Generally, the resolution of the CFA-ICP-TOFMS data is slightly better than that of the corresponding wet-chemistry measurement including the dispersion during the melting and in the respective debubbler volumes as a direct comparison of the high-frequency content of a section of the data has shown (Erhardt et al., 2019).

Figure 7 shows a 10-m section of the data at the full nominal 1-mm resolution. This example shows the very clear resolution
of the seasonal variability in all of the parameters as exploited in the revision of the GICC time scale (Sinnl et al., 2021). The peak dominating the electrolytic meltwater conductivity record in Figure 7 is a typical signal from aerosol depositon after a volcanic eruption. This specific eruption signal has previously been attributed to the 79 CE Vesuvius eruption which has recently been ruled out by tephra evidence pointing to a source in the Aleutian ark (Plunkett et al., 2022). In correspondence with other dating efforts (e.g. Sigl et al., 2015), the eruption is now dated to 88 CE, corresponding to 1910 yr b2k on the
GICC21 age scale (Sinnl et al., 2021).

For the depth resolution, the melt-speed and its stability plays an important role in determining the overall resolution as well as the stability of this resolution. Especially in the shallow parts of the core, above the depth when the firn reaches the density of ice at $917\,kg\,m^{-3}$, this is an important factor. For technical reasons, i.e., to supply enough meltwater for the system, the melt speed above the firn-ice transition needs to be increased to match the melt speed in ice-equivalent depth. This is clearly
visible in Figure 5E, with the exponential decrease of the melt speed for top $\sim 100\,m$ of the core. Through the adjustment of the melt speed, the data which is measured at constant time intervals is measured at a constant resolution in terms of the ice-equivalent depth. Assuming an approximate constant annual layer thickness, this translates to an approximate constant temporal resolution as well. Panel F in Figure 5 shows the variability of the melt speed within each of the measurement runs. This is a measure of how constant the resolution of the data is: high melt speed variability will translate directly into variability
of the measurement resolution in terms of depth along the core. Overall, this variability given by the relative standard deviation is significantly lower than 10 %, and mostly around 5 %. Runs with much higher variability are usually caused by sample quality issues which resulted in interrupted melting. In some cases such variability can be caused by too-thin ice sections that were intentionally melted faster to avoid contamination by intrusion of meltwater from the outside of the melthead to the clean



part inside. However, because of the mixing in the system and because the response time of detection method on the order of
~20 s, any short-term variability likely plays a minor role in terms of jitter or the determination of annual layer thicknesses
from the data.

## 5.4 Ca records

Because the Ca and $Ca^{2+}$ concentrations were measured using two different and entirely independently calibrated methods,
they allow for direct comparison between the fluorimetric detection and ICP-TOFMS measurements at high resolution. Figure 6
shows a direct comparison of the records at 10 cm resolution alongside a linear regression between the two, their correlation,
root mean square (rms) difference and average ratio. Generally both data sets agree very well with each other despite the
differences in measurement techniques as indicated by the low rms difference, high correlation and close to unity average
ratio. The linear regression between the two indicates a small constant offset between the two. This is likely caused by the
aforementioned $Ca^{2+}$ sensitivity and LOD issues in the top part of the data set as the offset vanishes when looking only at data
below 322.3 m. However, measurements below the baseline for $Ca^{2+}$ were removed from the data set and the apparent offset
is small in comparison to the concentrations and their uncertainties.

At higher resolution, as illustrated in Figure 7, the agreement between the two measurement is also very good. Focusing
on 1-m sections of the full resolution 1-mm data, the average correlation between the two measurements is 0.95 with an rms
difference of 2.44 ppb.

In summary, the agreement both at low and high resolution between the two data sets illustrates their compatibility as well as
their high quality. However, it is worth noting that it is not given a priori that the two should agree as the fundamentally measure
two different things. $Ca^{2+}$ in the meltwater steam is the result of water-soluble calcium species that are dissolved during the
melting and transport of the sample. These are only a subset of the total amount of Ca that is present in the meltwater which
is what the ICP-TOFMS measures. For example, in case of the presence of a high load of poorly soluble Ca-bearing mineral
dust particles in the ice the ICP-TOFMS will detect significantly more Ca than the fluorimetric $Ca^{2+}$ measurement. That being
said, the close agreement between the two suggests that most of the total Ca present in the ice is soluble in the meltwater
stream. A detailed comparison of the differences between the two and how they arise is planned.

For the drilling of the EGRIP ice core the same drill liquid combination of COASOAL and ESTISOL 240 (Popp et al., 2014)
was used as for the NEEM ice core. The drill liquid acts as an absorber of bivalent cations in the CFA system if it is introduced
by contaminated ice. This can have a detrimental effect on the measurement of $Ca^{2+}$ and likely all other bivalent cations by
producing strong absorption/desorption signals during measurement runs (Erhardt et al., 2022a). As almost all of the Calcium
detected by the ICP-TOFMS is also dissolved $Ca^{2+}$, the same is true for the total Ca measurements. To counteract the risk
of drill liquid entering the melt system, special care was taken during sample preparation to remove all possibly contaminated
ice. This is especially important for small, refrozen and often hard to see cracks in the sample. For the data presented here, this
effort, possibly in combination with the low Ca concentrations in the Holocene ice, has proven to be very fruitful. The data was
found to be free of the previously described absorption/desorption effects and no correction was necessary.



## 5.5 Possible upstream effects

As mentioned above, the EGRIP ice core is located inside of the Northeast Greenland Ice Stream and at the coring location surface velocities are $55\,\mathrm{m\,a^{-1}}$ (Hvidberg et al., 2020) and ice down-core originates from a location closer to the ice divide, likely characterized by higher accumulation rate (Gerber et al., 2021). This change in location and especially in the accumulation rate of about $\sim 30\,\%$ likely also has an influence on the aerosol data presented here both due to the change in deposition characteristics as well as due to the change in distance from possible source areas (e.g. Fischer et al., 1998a, b). Furthermore, short-term variability in the data sets is likely affected by the varying influence of the local topography changes upstream (e.g. Winski et al., 2019), leading for example to accumulation variability independent of precipitation on length scales of meters (sastrugies) to kilometers (dunes) due to wind reworking. That means, that both long-term trends as well as short-term variability in the data will need to be interpreted with great care and in the light of the influence of the upstream effects.

## 6 Conclusions

Here were present a multi-proxy record of impurity concentrations from the EGRIP ice core covering the past $3.8\,\mathrm{ka}$. The data was measured using the state-of-the-art continuous flow analysis system at the University of Bern and includes measurements of dissolved calcium, ammonium and nitrate from spectrophotometric measurements, electrolytic meltwater conductivity as well as total calcium and sodium concentrations measured by online ICP-TOFMS. The high-resolution records presented have sub-annual resolution and have formed part of the data base for the revised Greenland ice-core chronology, the GICC21, over the past $3.8\,\mathrm{ka}$ (Sinnl et al., 2021).

Summarizing from the previous section we stress the following points that should be considered when using this data:

1. The data sets have undergone extensive quality control both manually and automated. However, we can not guarantee the complete absence of any spurious signals, especially at the full resolution.

2. Similar to previous CFA data sets, the depth assignment to a given down-core depth is likely accurate to better than $1\,\mathrm{cm}$. The relative depth assignment within the CFA data set is much more accurate and uncertainties are on the order of $1\,\mathrm{mm}$.

3. Analytical uncertainties for the wet-chemistry data ($\mathrm{Ca^{2+}}$, $\mathrm{NH_4^+}$, $\mathrm{NO_3^-}$) is typically better than $10\,\%$, for the ICP-TOFMS data (Ca, Na) better than $5\,\%$.

4. The fluorescence spectrometric $\mathrm{Ca^{2+}}$ data above $322.3\,\mathrm{m}$ depth suffers from decreased sensitivity of the measurement setup and has higher-than-usual limits of detection. Values below the baseline were removed from the data set.

5. All the aerosol concentrations in the EGRIP ice core, including the ones presented here, are likely affected by upstream effects due to changes in the local deposition regime and distance from the aerosol source regions further upstream. These effects are subject to ongoing investigation and they need to be taken into account when interpreting the data at any temporal scale.



6. Last but not least, we stress that any detailed interpretation, especially of short-term signals in the data sets should always be confirmed at least within the multi-parameter data series, if not with additional data or measurements.

We invite any future user of these data sets to actively involve CFA or ice-core specialists into their investigation for an
expert assessment of the data quality/limitations in which may be specific to the application intended by the data user in order to avoid any shortcomings and pitfalls of the complex data sets.

## 7   Data availability

Both the full 1-mm-resolution as well as the decadal averaged data are available on PANGAEA under CC BY license (https://doi.pangaea.de/10.1594/PANGAEA.945293 Erhardt et al., 2022b):

– Full resolution: https://doi.org/10.1594/PANGAEA.945290

    – Decadal averages on the GICC21 age scale: https://doi.org/10.1594/PANGAEA.945291

The GICC21 age scale can be found in the supplement of Sinnl et al. (2021).

*Author contributions.* The CFA campaigns at the University of Bern were organized and hosted by TE and CMJ under the lead of TE. FA supported the wet-chemistry CFA measurements. The wet-chemistry CFA data was processed and quality controlled by CMJ, the ICP-
TOFMS data by TE. TE wrote the initial draft of the manuscript and prepared the figures. The manuscript and figures were finalized with input from HF, CMJ, HAK, MB, DS, AS, NM, RM, FB, FA and IC. All authors took part in the measurement campaigns or contributed to their success.

*Competing interests.* The authors declare no competing interests.

*Acknowledgements.* Purchase of the icpTOF has been made possible through the Swiss National Science Foundation (SNF) R'Equip pro-
posal iceCP-TOF (grant no. 206021_170739) and specific institutional funds made available by the University of Bern. The long-term financial support of ice core research at the University of Bern by the Swiss National Science Foundation (grant no. 200020_172506 (iCEP) and 20FI21_164190 (EGRIP)) is gratefully acknowledged.

    EGRIP is directed and organized by the Centre for Ice and Climate at the Niels Bohr Institute, University of Copenhagen. It is supported by funding agencies and institutions in Denmark (A. P. Møller Foundation, University of Copenhagen), USA (US National Science Foundation,
Office of Polar Programs), Germany (Alfred Wegener Institute, Helmholtz Centre for Polar and Marine Research), Japan (National Institute of Polar Research and Arctic Challenge for Sustainability), Norway (University of Bergen and Trond Mohn Foundation), Switzerland (Swiss National Science Foundation), France (French Polar Institute Paul-Emile Victor, Institute for Geosciences and Environmental research), Canada (University of Manitoba) and China (Chinese Academy of Sciences and Beijing Normal University).



MH, KF and OJ acknowledge financial support by JSPS KAKENHI (grant number JP18H04140).

The authors gratefully acknowledge the contributions of the countless people that facilitated and took part in the field campaigns, the ice-core drilling and processing as well helped during the CFA melting campaigns.

Furthermore the authors are especially grateful for the help of Priska Lehmann, Sam Black, Patrick Zens, Miriam Läderach, and Michelle Shu-Ting Lee during the campaigns.



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

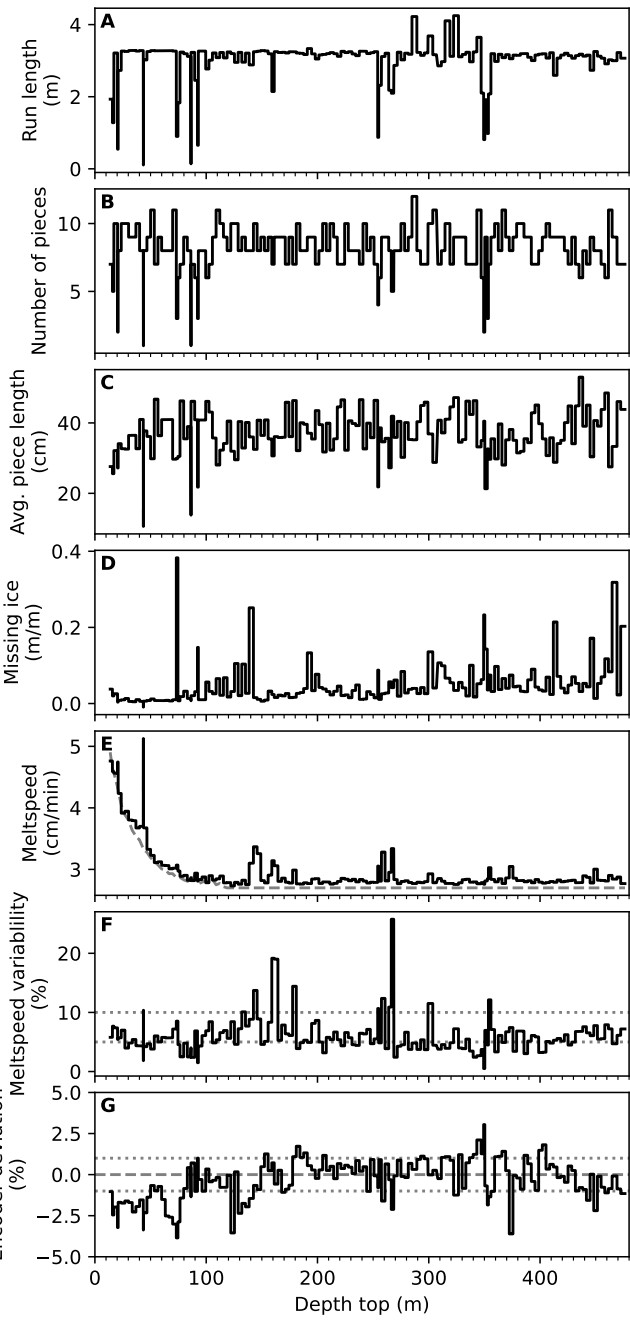

**Figure 5.** Depth assignment statistics. Panels A-D show information on the amount of ice melted during each run (A) and the number of pieces of ice (B), their average length and the amount of ice that is missing as indicators for the sample quality. Panels E-G show the melt speed (E) alongside the target meltspeed of $2.7\,\mathrm{cm\,min^{-1}}$ in ice equivalent, the interrun variability of the melt speed (F) as an indicator for the meltspeed stability and the encoder deviation (G) as an indicator of the depth-assignment precision. In panel F, the horizontal lines mark 5 and $10\,\%$ variability, in Panel G 0 and $\pm 1\,\%$ deviation.

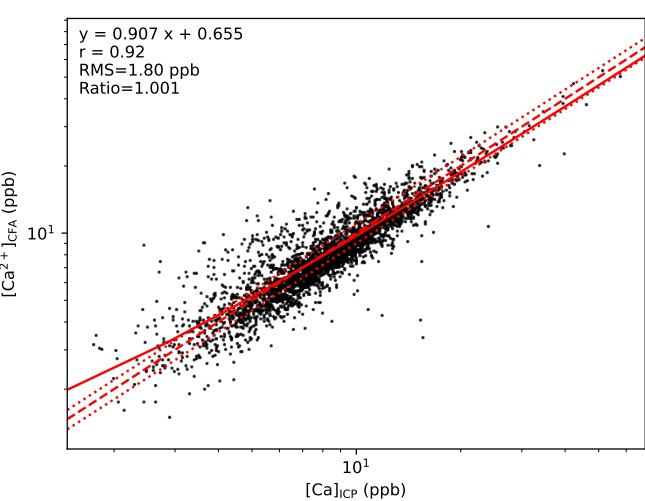

**Figure 6.** Cross plot of ICP-TOFMS Ca and fluorimetric $Ca^{2+}$ concentrations at $10\,cm$ resolution, considering only sections with less than $50\,\%$ missing data. Regression, Pearson correlation (r), root mean square error (rms) and average ratio were calculated on the concentrations (i.e. not log-transformed). The solid line is the result of the regression, the dashed line marks the 1:1 line, the dotted lines indicate a $\pm10\,\%$ deviation from that line.



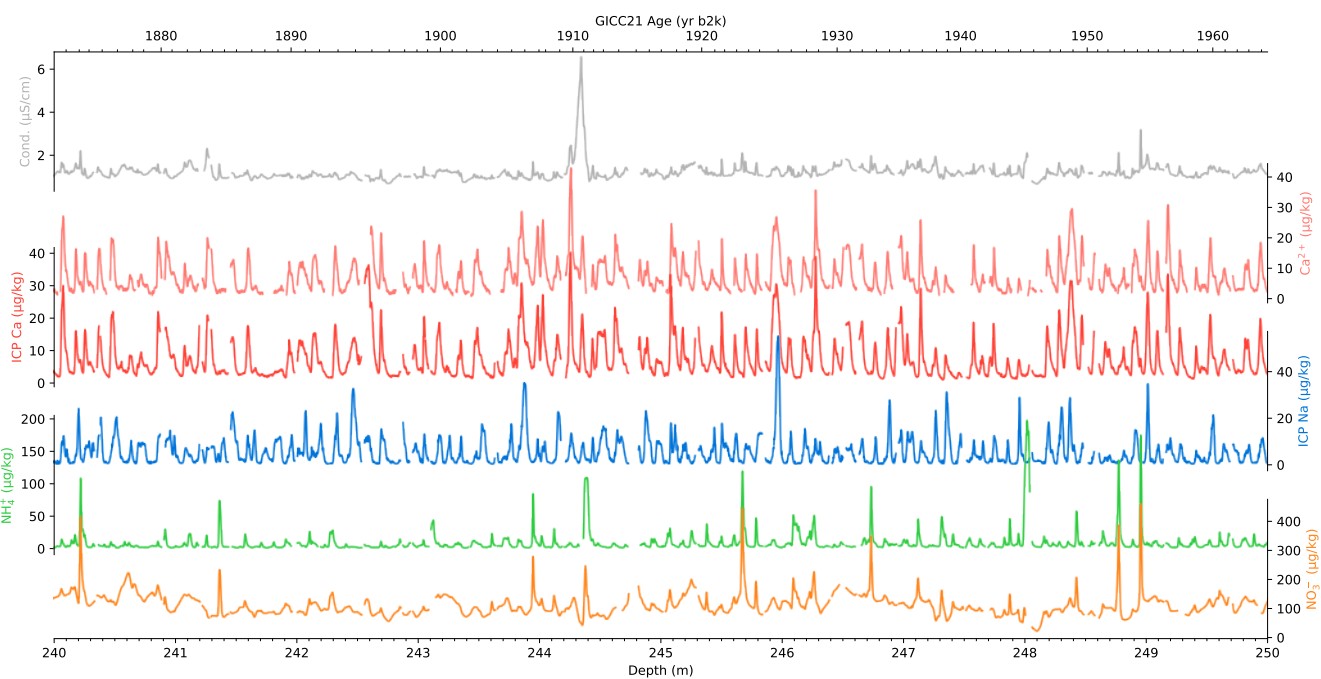

**Figure 7.** Example of high-resolution data over a 10-m section of the core. The parameters shown are the same as in Figure 3, however, data is shown on the depth scale, with the secondary x-axis at the top indicating the age at annual resolution. The large peak in conductivity at 244.3 m is a typical signal of a volcanic eruption, and has been used to synchronize the age scales between different ice cores.