# Peer review of "High resolution aerosol data from the top 3.8 ka of the EGRIP ice core"

_Earth System Science Data, 2023_

## Author Comment (AC1)

We thank both reviewers for their efforts to review the paper, for their kind words and for the excelent suggestions that they provided to improve the manuscipt. In the following we will adress all individual comments point by point.

**Response to Review Comment 1**

*This paper describes a dataset, deposited at Pangaea, that contains high-resolution (1 mm) CFA data for ions (and conductivity) from EGRIP. The data extend to 3800 years (479 m); they have been used to develop an age model so far; they are clearly supporting data for that and therefore have value, as well as future value for assessing changes in environmental factors over this time period. Much of the paper itself is devoted to describing methodology. This includes more general methodology about the drilling and CFA campaign (which also generated other data not shown here). The main novelty in the methodology compared to previous data papers concerns the ICP-TOFMS. The methodology seems clear and the data are undoubtedly of high quality, so the paper seems very publishable. I note that this is only about half the data that were measured in 2018 and 2019. While I understand why the data are being released in increments, it will be annoying for users if they remain in separate files, so I suggest that future releases are added to existing files rather than listed as separate files that readers have to splice together.*

Thanks. We plan to release the remaining data in files that are either including the data presented here or are at least easily combined with it.

**Minor comments**

*Lines 43, 51: accumulation rates are in cm/a – is this water equivalent or ice equivalent?*

Thanks for pointing this out. The values are ice-equivalent and we have added that to the text.

*Fig 2 caption. Sorry to be fussy but the correct abbreviation is mL not ml. Similarly uL later.*

We kindly disagree with the reviewer: both the capital L and the small l as abbreviation for the unit liter are accepted, for example by IUPAC, the standard followed by Copernicus Publications. We therefor are going to follow the recommendation of the editor in this respect.

*Line 83 "samples at 5——2.5 cm depth" – please explain what you mean by this. Do you mean "between 2.5 and 5 cm"? It's not obvious.*

Rephrased to be more explicit.

"Furthermore, samples at 5 cm and 2.5 cm depth resolution for ion chromatography (run by the Alfred Wegener Institute, Bremerhaven, Germany) as well as multiple aliquots at 55-cm resolution were collected for various offline measurements run by the international partners."

*Lines 84, 86, "aliquots" spelling*

Corrected

*Line 113 "Note, that the air is released from the ice during melting and thus does not pose a contamination risk". I'm not sure what this means, why would the air pose a contamination risk?*

Slightly rephrased to be more explicit. Any contact with additional gasses, even clean gasses, poses an additional contamination risk. The air from the ice itself is what the impurities where exposed to during preservation, so it is not adding any additional impurities to the meltwater stream.

"The air-water mixture coming from the melthead is used to efficiently transport particulates to the instrument and to limit signal dispersion. The air is released from the ice during melting and thus does not pose a contamination risk as would other external gas sources."

*Fig 3, line 4 of caption "most"*

Fixed.

*Fig 3 – for comparison purposes it would be better if the y-scaling of the two Ca datasets was the same, which seems not to be the case at present.*

Done, also in the high-resolution data figure.

*Line 202 "mis-measurement" spelling*

Corrected.

*Section 5.2 and Fig 5. I found this section hard to follow. Please spell out what you mean by "encoder deviation" in 5G – it seems from the text as if it should be telling me about the "total discrepancy between the length of ice as measured by the encoder to the length of ice measured using sample preparation" but I don't understand what you mean by this "encoder deviation". Also if its 1% (typically) for a 3 m piece, how is the depth precise to 1 cm? I believe you that it is, but I'm not clear how I get that from Figure 5. The figure is also not well-designed if you aim to show that run length doesn't affect the depth precision, as it is impossible to follow the few shorter runs down to section G.*

Thank you for pointing this out. Your comment prompted us to completely rewrite the section and change the estimate to be more conservative to around $2\,\mathrm{cm}$ uncertainty over a $3.3\,\mathrm{cm}$ measurement run which is still in line with the approximately $1\,\%$ relative uncertainty. We hope the new section is better understandable.

[revised manuscript text omitted]

*Line 239. Figure 7 is called before Figure 6. You may want to renumber them?*

This is an artifact of the figure placement and the resulting automatic numbering in the draft version of the paper. We will try to assure the correct numbering in the final version.

*Line 241 "deposition" spelling*

Fixed.

*Line 244 "88CE, corresponding to 1910 yr b2k" – am I missing something? Surely 88CE is 1912 b2k?*

That is indeed correct. We have changed the sentence now such that is mirrors the exact wording in (Sinnl et al., 2022) and removed the b2k date.

"In correspondence with other dating efforts (e.g. Sigl et al., 2015), the eruption is now dated to 87/88 CE on the GICC21 age scale (Sinnl et al., 2022)."

*Line 303 "Here we…"*

Corrected.

**Response to Review Comment 2**

*Erhardt and others present a multi-species dataset of aerosol records measured from the EastGRIP ice core (top 479 meters). The record being discussed covers the past 3,800 years and has a temporal resolution on sub-annual scale. Overall, the authors did a good job in describing the samples, the analytical methods, and the product dataset. I only have a few number questions that revolve around the clarity of the manuscript and a handful of technical comments/suggestions. Once they are addressed, I will be happy to recommend the acceptance of this manuscript.*

Thanks.

*Line a: here you acknowledge that the 1-mm-resolution dataset is an oversampling of the true record and the actual resolution is probably 1-cm. It would be better if you could move this sentence to line 3, right after "as well as decadal averages of these profiles." Otherwise, people might start calculating the temporal resolution themselves and apparently overestimate the temporal resolution by an order of magnitude.*

That is an excellent suggestion. We have reordered the sentences in the abstract accordingly.

"The data consists of 1 mm-depth-resolution profiles of calcium, sodium, ammonium, nitrate and electrolytic conductivity as well as decadal averages of these profiles. The nominally 1-mm data represents an oversampling of the record as the true resolution is limited by the analytical setup to approximately 1 cm. Alongside the data we provide a description of the measurement setup, procedures, the relevant references for the specific methods as well as an assessment of the precision of the measurements, the sample to depth assignment and the depth and temporal resolution of the data set. The error of absolute depth assignment of the data may be on the order of 1 cm, however relative depth offsets between the records of the individual species is only on the order of 1 mm. The presented data has sub-annual resolution over the entire depth range and has already formed part of the data for an annually layer-counted time scale for the EGRIP ice core used to improve and revise the multi-core Greenland ice-core chronology (GICC05) to a new version, GICC21 (Sinnl et al., 2022)."

*Line 10: there should be a blank in front of "GICC21".*

Fixed.

*Line 23: 'gold standard' is a pretty strong statement. Can you add some citations to this sentence to back it up?*

Change to 'de facto standard', which we think is adequate given that it has been applied by many groups to many of the deep and many of the shallow ice cores drilled since its inception.

"These setups have overcome some of the analytical challenges/limitations that the very low impurity concentrations and the desired high depth resolution pose and have become the de facto standard for long, high-resolution aerosol records."

*Line 44-45: it would be nice to note that the drilling is still ongoing as of writing (btw I enjoy reading the field diary!).*

So did we and the great news of the bedrock being reached in July 2023 was added to the text.

*Line 51: can you add a very quick description how Gerber et al (2021) derived those numbers (111 km further upstream and 160 m higher)?*

Added some more information to the text.

"An inversion of a two-dimensional flow model along different flow-lines to EGRIP using radar-stratigrahic horizons shows, that at 3.8 ka, the ice likely originated approximately 111 km further upstream at an elevation about 160 m higher than the coring location and closer to the ice divide (Gerber et al., 2021)."

*Line 58 (as well as line 97 below): the description here seems to suggest that the introduction of the ICP-TOFMS occurred between 2018 and 2019. Yet, looking at the data it appears that the ICP-TOFMS has been there all along; it's just the absorption method for Na+ was discontinued in 2019 and the data produced in this method were not included in the PANGEA dataset. Is this the case? If so, you might want to re-word some sections or, if space permits, have a separate table documenting what methods measured what properties in what year (or in what depth range), and whether those data are included in your dataset.*

We have clarified the relevant sentence in the paragraph (see below). We have also debated the addition of a complete table of of analytes that where measured but have decided against that as the data evaluation is still ongoing and the exact analytes that are of good quality are still not determined.

*Line 83: the '5 − −2.5 cm' expression seems strange and not very clear. Please consider rephrasing.*

Rephrased to be more explicit.

"Furthermore, samples at $5\,\mathrm{cm}$ and $2.5\,\mathrm{cm}$ depth resolution for ion chromatography (run by the Alfred Wegener Institute, Bremerhaven, Germany) as well as multiple aliquots at 55-cm resolution were collected for various offline measurements run by the international partners."

*Line 97 and onward: I recommend you discuss ICP-TOFMS exclusively in section 4.2.*

Removed or clarified mention of the ICP-TOFMS in section 4.1.

"In 2019 (starting from $350.35\,\mathrm{m}$) the absorption spectroscopic $Na^+$ measurement was dropped in favor of the more reliable and higher resolution measurement by the ICP-TOFMS introduced prior to the 2018 melting and discussed below (Erhardt et al., 2019). Furthermore, $H_2O_2$ measurements were halted after the 2018 season because of the loss of a usefill signal due to excessive diffusion. Here we only provide data of $Ca^{2+}$, $NH_4^+$ and $NO_3^-$ concentrations, electrolytic meltwater conductivity from the wet-chemistry CFA."

*Line 140: here you mentioned the presence of large gaps in the Ca2+ data but didn't explain why (except briefing doing so in the caption of Figure 3 and explaining the LOD issue in line 179). It would be better if you could succinctly describe the analytical issues that led to those gaps here so readers will better understand the possible limitations of the dataset.*

We wanted to make the point that the ICP-TOFMS dataset itself is not affected by these issues. We have added a reference to the analytical precision section of the text so the reader knows that there is more detail coming later on.

"Thus the depth assignment of the Ca data is well constrained even when calibration issues affects the $Ca^{2+}$ data as described in Section 5.4."

*Line 288: could you please elaborate how the potentially contaminated ice is removed, like by extensive trimming perhaps?*

Indeed this was either done by trimming or cutting out affected sections. We added that to the text.

"To counteract the risk of drill liquid entering the melt system, special care was taken during sample preparation to remove all possibly contaminated ice either by scraping off or cutting out possibly affected ice."

*Line 292: this section (5.5) doesn't seem to affect data quality directly but concerns the interpretation of the data, which is not what this manuscript aims to do. Maybe this section can be moved towards the beginning of the manuscript (in section 2 for example).*

Good point that the upstream effects are not part of the data quality. Instead of moving the section to the beginning of the manuscript, we instead elected to make this an extra section at the end of the manuscript.